# PAC-Bayesian Bound for the Conditional Value at Risk

**Zakaria Mhammedi**
The Australian National University and Data61
zak.mhammedi@anu.edu.au

**Benjamin Guedj**
Inria and University College London
benjamin.guedj@inria.fr

**Robert C. Williamson**
bobwilliamsonoz@icloud.com

## Abstract

Conditional Value at Risk (CVAR) is a family of "coherent risk measures" which generalize the traditional mathematical expectation. Widely used in mathematical finance, it is garnering increasing interest in machine learning, e.g., as an alternate approach to regularization, and as a means for ensuring fairness. This paper presents a generalization bound for learning algorithms that minimize the CVAR of the empirical loss. The bound is of PAC-Bayesian type and is guaranteed to be small when the empirical CVAR is small. We achieve this by reducing the problem of estimating CVAR to that of merely estimating an expectation. This then enables us, as a by-product, to obtain concentration inequalities for CVAR even when the random variable in question is unbounded.

## 1 Introduction

The goal in statistical learning is to learn hypotheses that generalize well, which is typically formalized by seeking to minimize the *expected* risk associated with a given loss function. In general, a loss function is a map $\ell \colon \mathcal{H} \times \mathcal{X} \to \mathbb{R}_{\geq 0}$, where $\mathcal{X}$ is a feature space and $\mathcal{H}$ is an hypotheses space. In this case, the *expected risk* associated with a given hypothesis $h \in \mathcal{H}$ is given by $\mathrm{R}[\ell(h, X)] \coloneqq \mathrm{E}[\ell(h, X)]$. Since the data-generating distribution is typically unknown, the expected risk is approximated using observed i.i.d. samples $X_1, \ldots, X_n$ of $X$, and an hypothesis is then chosen to minimize the empirical risk $\widehat{\mathrm{R}}[\ell(h, X)] \coloneqq \sum_{i=1}^{n} \ell(h, X_i)/n$. When choosing an hypothesis $\hat{h}$ based on the empirical risk $\widehat{\mathrm{R}}$, one would like to know how close $\widehat{\mathrm{R}}[\ell(\hat{h}, X)]$ is to the actual risk $\mathrm{R}[\ell(\hat{h}, X)]$; only then can one infer something about the *generalization* property of the learned hypothesis $\hat{h}$.

Generalization bounds—in which the expected risk is bounded in terms of its empirical version up to some error—are at the heart of many machine learning problems. The main techniques leading to such bounds comprise uniform convergence arguments (often involving the Rademacher complexity of the set $\mathcal{H}$), algorithmic stability arguments (see e.g. [Bousquet and Elisseeff, 2002] and more recently the work from [Abou-Moustafa and Szepesvári, 2019, Bousquet et al., 2019, Celisse and Guedj, 2016]), and the PAC-Bayesian analysis for non-degenerate randomized estimators [McAllester, 2003]. Behind these techniques lies *concentration inequalities*, such as Chernoff's inequality (for the PAC-Bayesian analysis) and McDiarmid's inequality (for algorithmic stability and the uniform convergence analysis), which control the deviation between population and empirical averages [see Boucheron et al., 2003, 2013, McDiarmid, 1998, among others].

Standard concentration inequalities are well suited for learning problems where the goal is to minimize the expected risk $\mathrm{E}[\ell(h, X)]$. However, the expected risk—the mean performance of an algorithm—might fail to capture the underlying phenomenon of interest. For example, when

dealing with medical (responsivity to a specific drug with grave side effects, etc.), environmental (such as pollution, exposure to toxic compounds, etc.), or sensitive engineering tasks (trajectory evaluation for autonomous vehicles, etc.), the mean performance is not necessarily the best objective to optimize as it will cover potentially disastrous mistakes (*e.g.*, a few extra centimeters when crossing another vehicle, a slightly too large dose of a lethal compound, etc.) while possibly improving on average. There is thus a growing interest to work with alternative measures of risk (other than the expectation) for which standard concentration inequalities do not apply directly. Of special interest are *coherent risk measures* [Artzner et al., 1999] which possess properties that make them desirable in mathematical finance and portfolio optimization [Allais, 1953, Ellsberg, 1961, Rockafellar et al., 2000], with a focus on optimizing for the worst outcomes rather than on average. Coherent risk measures have also been recently connected to fairness, and appear as a promising framework to control the fairness of an algorithm's solution [Williamson and Menon, 2019].

A popular coherent risk measure is the Conditional Value at Risk (CVAR; see Pflug, 2000); for $\alpha \in (0,1)$ and random variable $Z$, $\text{CVAR}_\alpha[Z]$ measures the expectation of $Z$ conditioned on the event that $Z$ is greater than its $(1-\alpha)$-th quantile. CVAR has been shown to underlie the classical SVM [Takeda and Sugiyama, 2008], and has in general attracted a large interest in machine learning over the past two decades [Bhat and Prashanth, 2019, Chen et al., 2009, Chow and Ghavamzadeh, 2014, Huo and Fu, 2017, Morimura et al., 2010, Pinto et al., 2017, Prashanth and Ghavamzadeh, 2013, Takeda and Kanamori, 2009, Tamar et al., 2015, Williamson and Menon, 2019].

Various concentration inequalities have been derived for $\text{CVAR}_\alpha[Z]$, under different assumptions on $Z$, which bound the difference between $\text{CVAR}_\alpha[Z]$ and its standard estimator $\widehat{\text{CVAR}}_\alpha[Z]$ with high probability [Bhat and Prashanth, 2019, Brown, 2007, Kolla et al., 2019, Prashanth and Ghavamzadeh, 2013, Wang and Gao, 2010]. However, none of these works extend their results to the statistical learning setting where the goal is to learn an hypothesis from data to minimize the conditional value at risk. In this paper, we fill this gap by presenting a sharp PAC-Bayesian generalization bound when the objective is to minimize the conditional value at risk.

**Related Works.**   Deviation bounds for CVAR were first presented by Brown [2007]. However, their approach only applies to bounded continuous random variables, and their lower deviation bound has a sub-optimal dependence on the level $\alpha$. Wang and Gao [2010] later refined their analysis to recover the "correct" dependence in $\alpha$, albeit their technique still requires a two-sided bound on the random variable $Z$. Thomas and Learned-Miller [2019] derived new concentration inequalities for CVAR with a very sharp empirical performance, even though the dependence on $\alpha$ in their bound is sub-optimal. Further, they only require a one-sided bound on $Z$, without a continuity assumption.

Kolla et al. [2019] were the first to provide concentration bounds for CVAR when the random variable $Z$ is unbounded, but is either sub-Gaussian or sub-exponential. Bhat and Prashanth [2019] used a bound on the Wasserstein distance between true and empirical cumulative distribution functions to substantially tighten the bounds of Kolla et al. [2019] when $Z$ has finite exponential or $k$th-order moments; they also apply their results to other coherent risk measures. However, when instantiated with bounded random variables, their concentration inequalities have sub-optimal dependence in $\alpha$.

On the statistical learning side, Duchi and Namkoong [2018] present generalization bounds for a class of coherent risk measures that technically includes CVAR. However, their bounds are based on uniform convergence arguments which lead to looser bounds compared with ours. Another bound based on a uniform convergence argument was also presented in a concurrent work by Curi et al. [2020].

**Contributions.**   Given a learning algorithm that outputs a posterior distribution $\widehat{\rho}$ on $\mathcal{H}$, our main contribution is a *PAC-Bayesian generalization bound for the conditional value at risk*, where we bound the difference $\text{CVAR}_\alpha[Z] - \widehat{\text{CVAR}}_\alpha[Z]$, for $\alpha \in (0,1)$ and $Z \coloneqq \text{E}_{h \sim \widehat{\rho}}[\ell(h, X)]$, by a term of order $\sqrt{\widehat{\text{CVAR}}_\alpha[Z] \cdot \mathcal{K}_n / (n\alpha)}$, with $\mathcal{K}_n$ representing a complexity term which depends on $\mathcal{H}$. Due to the presence of $\widehat{\text{CVAR}}_\alpha[Z]$ inside the square-root, our generalization bound has the desirable property that it becomes small whenever the empirical conditional value at risk is small. For the standard expected risk, only state-of-the-art PAC-Bayesian bounds share this property (see *e.g.* Catoni [2007], Langford and Shawe-Taylor [2003], Maurer [2004] or more recently in Mhammedi et al. [2019], Tolstikhin and Seldin [2013]). We refer to [Guedj, 2019] for a recent survey on PAC-Bayes.

As a by-product of our analysis, we derive a new way of obtaining *concentration bounds for the conditional value at risk* by reducing the problem to estimating expectations using empirical means. This reduction then makes it easy to obtain concentration bounds for $\mathrm{CVAR}_\alpha[Z]$ even when the random variable $Z$ is unbounded ($Z$ may be sub-Gaussian or sub-exponential). Our bounds have explicit constants and are sharper than existing ones due to Bhat and Prashanth [2019], Kolla et al. [2019] which deal with the unbounded case.

**Outline.**    In Section 2, we define the conditional value at risk along with its standard estimator. In Section 3, we recall the statistical learning setting and present our PAC-Bayesian bound for CVAR. The proof of our main bound is in Section 4. In Section 5, we present a new way of deriving concentration bounds for CVAR which stems from our analysis in Section 4. Section 6 concludes and suggests future directions.

## 2   Preliminaries

Let $(\Omega, \mathcal{F}, P)$ be a probability space. For $p \in \mathbb{N}$, we denote by $\mathcal{L}^p(\Omega) \coloneqq \mathcal{L}^p(\Omega, \mathcal{F}, P)$ the space of $p$-integrable functions, and we let $\mathcal{M}_P(\Omega)$ be the set of probability measures on $\Omega$ which are absolutely continuous with respect to $P$. We reserve the notation E for the expectation under the reference measure $P$, although we sometimes write $\mathrm{E}_P$ for clarity. For random variables $Z_1, \ldots, Z_n$, we denote $\widehat{P}_n \coloneqq \sum_{i=1}^n \delta_{Z_i}/n$ the empirical distribution, and we let $Z_{1:n} \coloneqq (Z_1, \ldots, Z_n)$. Furthermore, we let $\pi \coloneqq (1, \ldots, 1)^\top/n \in \mathbb{R}^n$ be the uniform distribution on the simplex. Finally, we use the notation $\widetilde{\mathcal{O}}$ to hide log-factors in the sample size $n$.

**Coherent Risk Measures (CRM).**    A CRM [Artzner et al., 1999] is a functional $\mathrm{R}\colon \mathcal{L}^1(\Omega) \to \mathbb{R} \cup \{+\infty\}$ that is simultaneously, positive homogeneous, monotonic, translation equivariant, and sub-additive[1] (see Appendix B for a formal definition). For $\alpha \in (0, 1)$ and a real random variable $Z \in \mathcal{L}_1(\Omega)$, the conditional value at risk $\mathrm{CVAR}_\alpha[Z]$ is a CRM and is defined as the mean of the random variable $Z$ conditioned on the event that $Z$ is greater than its $(1-\alpha)$-th quantile[2]. This is equivalent to the following expression, which is more convenient for our analysis:

$$\mathrm{CVAR}_\alpha[Z] = \mathrm{C}_\alpha[Z] \coloneqq \inf_{\mu \in \mathbb{R}} \left\{ \mu + \frac{\mathrm{E}[Z-\mu]_+}{\alpha} \right\}.$$

Key to our analysis is the *dual representation* of CRMs. It is known that any CRM $\mathrm{R}\colon \mathcal{L}^1(\Omega) \to \mathbb{R}\cup\{+\infty\}$ can be expressed as the *support function* of some closed convex set $\mathcal{Q} \subseteq \mathcal{L}^1(\Omega)$ [Rockafellar and Uryasev, 2013]; that is, for any real random variable $Z \in \mathcal{L}^1(\Omega)$, we have

$$\mathrm{R}[Z] = \sup_{q \in \mathcal{Q}} \mathrm{E}_P[Zq] = \sup_{q \in \mathcal{Q}} \int_\Omega Z(\omega) q(\omega) \mathrm{d}P(\omega). \quad \text{(dual representation)} \tag{1}$$

In this case, the set $\mathcal{Q}$ is called the *risk envelope* associated with the risk measure R. The risk envelope $\mathcal{Q}_\alpha$ of $\mathrm{CVAR}_\alpha[Z]$ is given by

$$\mathcal{Q}_\alpha \coloneqq \left\{ q \in \mathcal{L}^1(\Omega) \;\middle|\; \exists Q \in \mathcal{M}_P(\Omega), \; q = \frac{\mathrm{d}Q}{\mathrm{d}P} \le \frac{1}{\alpha} \right\}, \tag{2}$$

and so substituting $\mathcal{Q}_\alpha$ for $\mathcal{Q}$ in (1) yields $\mathrm{CVAR}_\alpha[Z]$.[3] Though the overall approach we take in this paper may be generalizable to other popular CRMs, (see Appendix B) we focus our attention on CVAR for which we derive new PAC-Bayesian and concentration bounds in terms of its natural estimator $\widehat{\mathrm{CVAR}}_\alpha[Z]$; given i.i.d. copies of $Z_1, \ldots, Z_n$ of $Z$, we define

$$\widehat{\mathrm{CVAR}}_\alpha[Z] \coloneqq \widehat{\mathrm{C}}_\alpha[Z] \coloneqq \inf_{\mu \in \mathbb{R}} \left\{ \mu + \sum_{i=1}^n \frac{[Z_i - \mu]_+}{n\alpha} \right\}. \tag{3}$$

From now on, we write $\mathrm{C}_\alpha[Z]$ and $\widehat{\mathrm{C}}_\alpha[Z]$ for $\mathrm{CVAR}_\alpha[Z]$ and $\widehat{\mathrm{CVAR}}_\alpha[Z]$, respectively.

# 3 PAC-Bayesian Bound for the Conditional Value at Risk

In this section, we briefly describe the statistical learning setting, formulate our goal, and present our main results.

In the statistical learning setting, $Z$ is a *loss* random variable which can be written as $Z = \ell(h, X)$, where $\ell \colon \mathcal{H} \times \mathcal{X} \to \mathbb{R}_{\geq 0}$ is a loss function and $\mathcal{X}$ [resp. $\mathcal{H}$] is a feature [resp. hypotheses] space. The aim is to learn an hypothesis $\hat{h} = \hat{h}(X_{1:n}) \in \mathcal{H}$, or more generally a distribution $\hat{\rho} = \hat{\rho}(X_{1:n})$ over $\mathcal{H}$ (also referred to as randomized estimator), based on i.i.d. samples $X_1, \ldots, X_n$ of $X$ which minimizes some measure of risk—typically, the expected risk $\mathrm{E}_P[\ell(\hat{\rho}, X)]$, where $\ell(\hat{\rho}, X) \coloneqq \mathrm{E}_{h \sim \hat{\rho}}[\ell(h, X)]$.

Our work is motivated by the idea of replacing this expected risk by any coherent risk measure R. In particular, if $\mathcal{Q}$ is the risk envelope associated with R, then our quantity of interest is

$$\mathrm{R}[\ell(\hat{\rho}, X)] \coloneqq \sup_{q \in \mathcal{Q}} \int_\Omega \ell(\hat{\rho}, X(\omega)) q(\omega) \mathrm{d}P(\omega).$$

Thus, given a consistent estimator $\widehat{\mathrm{R}}[\ell(\hat{\rho}, X)]$ of $\mathrm{R}[\ell(\hat{\rho}, X)]$ and some prior distribution $\rho_0$ on $\mathcal{H}$, our grand goal (which goes beyond the scope of this paper) is to bound the risk $\mathrm{R}[\ell(\hat{\rho}, X)]$ as

$$\mathrm{R}[\ell(\hat{\rho}, X)] \leq \widehat{\mathrm{R}}[\ell(\hat{\rho}, X)] + \widetilde{\mathcal{O}}\left(\sqrt{\frac{\mathrm{KL}(\hat{\rho} \| \rho_0)}{n}}\right), \tag{4}$$

with high probability. Based on (3), the consistent estimator we use for $\mathrm{C}_\alpha[\ell(\hat{\rho}, X)]$ is

$$\widehat{\mathrm{C}}_\alpha[\ell(\hat{\rho}, X)] \coloneqq \inf_{\mu \in \mathbb{R}} \left\{ \mu + \sum_{i=1}^n \frac{[\ell(\hat{\rho}, X_i) - \mu]_+}{n\alpha} \right\}, \quad \alpha \in (0, 1). \tag{5}$$

This is in fact a consistent estimator (see e.g. [Duchi and Namkoong, 2018, Proposition 9]). As a first step towards the goal in (4), we derive a sharp PAC-Bayesian bound for the conditional value at risk, which we state now as our main theorem:

**Theorem 1.** *Let $\alpha \in (0, 1)$, $\delta \in (0, 1/2)$, $n \geq 2$, and $N \coloneqq \lceil \log_2(n/\alpha) \rceil$. Further, let $\rho_0$ be any distribution on an hypothesis set $\mathcal{H}$, $\ell \colon \mathcal{H} \times \mathcal{X} \to [0, 1]$ be a loss, and $X_1, \ldots, X_n$ be i.i.d. copies of $X$. Then, for any "posterior" distribution $\hat{\rho} = \hat{\rho}(X_{1:n})$ over $\mathcal{H}$, $\varepsilon_n \coloneqq \sqrt{\frac{2 \ln(N/\delta)}{\alpha n}} + \frac{\ln(N/\delta)}{3\alpha n}$, and $\mathcal{K}_n \coloneqq \mathrm{KL}(\hat{\rho} \| \rho_0) + \ln(N/\delta)$, we have, with probability at least $1 - 2\delta$.*

$$\mathrm{E}_{h \sim \hat{\rho}}[\mathrm{C}_\alpha[\ell(h, X)]] \leq \widehat{\mathrm{C}}_\alpha[\ell(\hat{\rho}, X)] + \sqrt{\frac{27 \widehat{\mathrm{C}}_\alpha[\ell(\hat{\rho}, X)] \mathcal{K}_n}{5\alpha n}} + 2\varepsilon_n \widehat{\mathrm{C}}_\alpha[\ell(\hat{\rho}, X)] + \frac{27 \mathcal{K}_n}{5n\alpha}. \tag{6}$$

**Discussion of the bound.**     Although we present the bound for the bounded loss case, our result easily generalizes to the case where $\ell(h, X)$ is sub-Gaussian or sub-exponential, for all $h \in \mathcal{H}$. We discuss this in Section 5. Our second observation is that since $\mathrm{C}_\alpha[Z]$ is a coherent risk measure, it is convex in $Z$ [Rockafellar and Uryasev, 2013], and so we can further bound the term $\mathrm{E}_{h \sim \hat{\rho}}[\mathrm{C}_\alpha[\ell(h, X)]]$ on the LHS of (6) from below by $\mathrm{C}_\alpha[\ell(\hat{\rho}, X)] = \mathrm{C}_\alpha[\mathrm{E}_{h \sim \hat{\rho}}[\ell(h, X)]]$. This shows that the type of guarantee we have in (6) is in general tighter than the one in (4).

Even though not explicitly done before, a PAC-Bayesian bound of the form (4) can be derived for a risk measure R using an existing technique due to McAllester [2003] as soon as, for any fixed hypothesis $h$, the difference $\mathrm{R}[\ell(h, X)] - \widehat{\mathrm{R}}[\ell(h, X)]$ is sub-exponential with a sufficiently fast tail decay as a function of $n$ (see the proof of Theorem 1 in [McAllester, 2003]). While it has been shown that the difference $\mathrm{C}_\alpha[Z] - \widehat{\mathrm{C}}_\alpha[Z]$ also satisfies this condition for bounded i.i.d. random variables $Z, Z_1, \ldots, Z_n$ (see *e.g.* Brown [2007], Wang and Gao [2010]), applying the technique of McAllester [2003] yields a bound on $\mathrm{E}_{h \sim \hat{\rho}}[\mathrm{C}_\alpha[\ell(h, X)]]$ (*i.e.* the LHS of (6)) of the form

$$\mathrm{E}_{h \sim \hat{\rho}}[\widehat{\mathrm{C}}_\alpha[\ell(h, X)]] + \sqrt{\frac{\mathrm{KL}(\hat{\rho} \| \rho_0) + \ln \frac{n}{\delta}}{\alpha n}}. \tag{7}$$

Such a bound is weaker than ours in two ways: **(I)** by Jensen's inequality the term $\widehat{\mathrm{C}}_\alpha[\ell(\hat{\rho}, X)]$ in our bound (defined in (5)) is always smaller than the term $\mathrm{E}_{h \sim \hat{\rho}}[\widehat{\mathrm{C}}_\alpha[\ell(h, X)]]$ in (7); and **(II)** unlike in our bound in (6), the complexity term inside the square-root in (7) does not multiply the

empirical conditional value at risk $\widehat{C}_\alpha[\ell(\widehat{\rho}, X)]$. This means that our bound can be much smaller whenever $\widehat{C}_\alpha[\ell(\widehat{\rho}, X)]$ is small—this is to be expected in the statistical learning setting since $\widehat{\rho}$ will typically be picked by an algorithm to minimize the empirical value $\widehat{C}_\alpha[\ell(\widehat{\rho}, X)]$. This type of improved PAC-Bayesian bound, where the empirical error appears multiplying the complexity term inside the square-root, has been derived for the expected risk in works such as [Catoni, 2007, Langford and Shawe-Taylor, 2003, Maurer, 2004, Seeger, 2002]; these are arguably the state-of-the-art generalization bounds.

**A reduction to the expected risk.** A key step in the proof of Theorem 1 is to show that for a real random variable $Z$ (not necessarily bounded) and $\alpha \in (0, 1)$, one can construct a function $g \colon \mathbb{R} \to \mathbb{R}$ such that the auxiliary variable $Y = g(Z)$ satisfies **(I)**

$$E[Y] = E[g(Z)] = C_\alpha[Z]; \tag{8}$$

and **(II)** for i.i.d. copies $Z_{1:n}$ of $Z$, the i.i.d. random variables $Y_1 \coloneqq g(Z_1), \ldots, Y_n \coloneqq g(Z_n)$ satisfy

$$\frac{1}{n} \sum_{i=1}^{n} Y_i \leq \widehat{C}_\alpha[Z](1 + \epsilon_n), \quad \text{where } \epsilon_n = \widetilde{\mathcal{O}}(\alpha^{-1/2} n^{-1/2}), \tag{9}$$

with high probability. Thus, due to (8) and (9), bounding the difference

$$E[Y] - \frac{1}{n} \sum_{i=1}^{n} Y_i, \tag{10}$$

is sufficient to obtain a concentration bound for CVAR. Since $Y_1, \ldots, Y_n$ are i.i.d., one can apply standard concentration inequalities, which are available whenever $Y$ is sub-Gaussian or sub-exponential, to bound the difference in (10). Further, we show that whenever $Z$ is sub-Gaussian or sub-exponential, then essentially so is $Y$. Thus, our method allows us to obtain concentration inequalities for $\widehat{C}_\alpha[Z]$, even when $Z$ is unbounded. We discuss this in Section 5.

## 4 Proof Sketch for Theorem 1

In this section, we present the key steps taken to prove the bound in Theorem 1. We organize the proof in three subsections. In Subsection 4.1, we introduce an auxiliary estimator $\widetilde{C}_\alpha[Z]$ for $C_\alpha[Z]$, $\alpha \in (0, 1)$, which will be useful in our analysis; in particular, we bound this estimator in terms of $\widehat{C}_\alpha[Z]$ (as in (9) above, but with the LHS replaced by $\widetilde{C}_\alpha[Z]$). In Subsection 4.2, we introduce an auxiliary random variable $Y$ whose expectation equals $C_\alpha[Z]$ (as in (8)) and whose empirical mean is bounded from above by the estimator $\widetilde{C}_\alpha[Z]$ introduced in Subsection 4.1—this enables the reduction described at the end of Section 3. In Subsection 4.3, we conclude the argument by applying the classical Donsker-Varadhan variational formula [Csiszár, 1975, Donsker and Varadhan, 1976].

### 4.1 An Auxiliary Estimator for CVAR

In this subsection, we introduce an auxiliary estimator $\widetilde{C}_\alpha[Z]$ of $C_\alpha[Z]$ and show that it is not much larger than $\widehat{C}_\alpha[Z]$. For $\alpha, \delta \in (0, 1)$, $n \in \mathbb{N}$, and $\pi \coloneqq (1, \ldots, 1)^\top / n \in \mathbb{R}^n$, define:

$$\widetilde{\mathcal{Q}}_\alpha \coloneqq \left\{ \boldsymbol{q} \in [0, 1/\alpha]^n \ : \ |E_{i \sim \pi}[q_i] - 1| \leq \epsilon_n \right\}, \quad \text{where} \quad \epsilon_n \coloneqq \sqrt{\frac{2 \ln \frac{1}{\delta}}{\alpha n}} + \frac{\ln \frac{1}{\delta}}{3 \alpha n}. \tag{11}$$

Using the set $\widetilde{\mathcal{Q}}_\alpha$, and given i.i.d. copies $Z_1, \ldots, Z_n$ of $Z$, let

$$\widetilde{C}_\alpha[Z] \coloneqq \sup_{\boldsymbol{q} \in \widetilde{\mathcal{Q}}_\alpha} \frac{1}{n} \sum_{i=1}^{n} Z_i q_i. \tag{12}$$

In the next lemma, we give a "variational formulation" of $\widetilde{C}_\alpha[Z]$, which will be key in our results:

**Lemma 2.** *Let $\alpha, \delta \in (0, 1)$, $n \in \mathbb{N}$, and $\widetilde{C}_\alpha[Z]$ be as in (12). Then, for any $Z_1, \ldots, Z_n \in \mathbb{R}$,*

$$\widetilde{C}_\alpha[Z] = \inf_{\mu \in \mathbb{R}} \left\{ \mu + |\mu| \epsilon_n + \frac{E_{i \sim \pi}[Z_i - \mu]_+}{\alpha} \right\}, \quad \text{where } \epsilon_n \text{ as in (11).} \tag{13}$$

The proof of Lemma 2 (which is in Appendix A.1) is similar to that of the generalized Donsker-Varadhan variational formula considered in [Beck and Teboulle, 2003]. The "variational formulation" on the RHS of (13) reveals some similarity between $\widetilde{C}_\alpha[Z]$ and the standard estimator $\widehat{C}_\alpha[Z]$ defined in (3). In fact, thanks to Lemma 2, we have the following relationship between the two:

**Lemma 3.** *Let $\alpha, \delta \in (0,1)$, $n \in \mathbb{N}$, and $Z_1, \ldots, Z_n \in \mathbb{R}_{\geq 0}$. Further, let $Z_{(1)}, \ldots, Z_{(n)}$ be the decreasing order statistics of $Z_1, \ldots, Z_n$. Then, for $\epsilon_n$ as in (11), we have*

$$\widetilde{C}_\alpha[Z] \leq \widehat{C}_\alpha[Z] \cdot (1 + \epsilon_n); \tag{14}$$

*and if $Z_1, \ldots, Z_n \in \mathbb{R}$ (not necessarily positive), then*

$$\widetilde{C}_\alpha[Z] \leq \widehat{C}_\alpha[Z] + |Z_{(\lceil n\alpha \rceil)}| \cdot \epsilon_n. \tag{15}$$

The inequality in (15) will only be relevant to us in the case where $Z$ maybe negative, which we deal with in Section 5 when we derive new concentration bounds for CVAR.

## 4.2 An Auxiliary Random Variable

In this subsection, we introduce a random variable $Y$ which satisfies the properties in (8) and (9), where $Y_1, \ldots, Y_n$ are i.i.d. copies of $Y$ (this is where we leverage the dual representation in (1)). This allows us to the reduce the problem of estimating CVAR to that of estimating an expectation.

Let $\mathcal{X}$ be an arbitrary set, and $f \colon \mathcal{X} \to \mathbb{R}$ be some fixed measurable function (we will later set $f$ to a specific function depending on whether we want a new concentration inequality or a PAC-Bayesian bound for CVAR). Given a random variable $X$ in $\mathcal{X}$ (arbitrary for now), we define

$$Z := f(X) \tag{16}$$

and the auxiliary random variable:

$$Y := Z \cdot \mathrm{E}[q_\star \mid X] = f(X) \cdot \mathrm{E}[q_\star \mid X], \quad \text{where} \quad q_\star \in \underset{q \in \mathcal{Q}_\alpha}{\operatorname{argmax}} \, \mathrm{E}[Zq], \tag{17}$$

and $\mathcal{Q}_\alpha$ as in (2). In the next lemma, we show two crucial properties of the random variable $Y$—these will enable the reduction mentioned at the end of Section 3:

**Lemma 4.** *Let $\alpha, \delta \in (0,1)$ and $X_1, \ldots, X_n$ be i.i.d. random variables in $\mathcal{X}$. Then, (I) the random variable $Y$ in (17) and $Y_i := Z_i \cdot \mathrm{E}[q_\star \mid X_i], i \in [n]$, where $Z_i := f(X_i)$, are i.i.d. and satisfy $\mathrm{E}[Y] = \mathrm{E}[Y_i] = C_\alpha[Z]$, for all $i \in [n]$; and (II) with probability at least $1 - \delta$,*

$$(\mathrm{E}[q_\star \mid X_1], \ldots, \mathrm{E}[q_\star \mid X_n])^\top \in \widetilde{Q}_\alpha, \quad \text{where } \widetilde{Q}_\alpha \text{ is as in (11)}. \tag{18}$$

The random variable $Y$ introduced in (17) will now be useful since due to (18) in Lemma 4, we have, for $\alpha, \delta \in (0,1)$; $Z$ as in (16); and i.i.d. random variables $X, X_1, \ldots, X_n \in \mathcal{X}$,

$$P\left[\frac{1}{n} \sum_{i=1}^n Y_i \leq \widetilde{C}_\alpha[Z]\right] \geq 1 - \delta, \quad \text{where } Y_i = Z_i \cdot \mathrm{E}[q_\star \mid X_i] \tag{19}$$

and $\widetilde{C}_\alpha[Z]$ as in (12). We now present a concentration inequality for the random variable $Y$ in (17); the proof, which can be found in Appendix A, is based on a version of the standard Bernstein's moment inequality [Cesa-Bianchi and Lugosi, 2006, Lemma A.5]:

**Lemma 5.** *Let $X, (X_i)_{i \in [n]}$ be i.i.d. random variables in $\mathcal{X}$. Further, let $Y$ be as in (17), and $Y_i = f(X_i) \cdot \mathrm{E}[q_\star \mid X_i], i \in [n]$, with $q_\star$ as in (17). If $\{f(x) \mid x \in \mathcal{X}\} \subseteq [0,1]$, then for all $\eta \in [0, \alpha]$,*

$$\mathrm{E}\left[\exp\left(n\eta \cdot \left(\mathrm{E}_P[Y] - \frac{1}{n} \sum_{i=1}^n Y_i - \frac{\eta \kappa(\eta/\alpha)}{\alpha} C_\alpha[Z]\right)\right)\right] \leq 1, \quad \text{where } Z = f(X),$$

*and $\kappa(x) := (e^x - 1 - x)/x^2$, for $x \in \mathbb{R}$.*

Lemma 5 will be our starting point for deriving the PAC-Bayesian bound in Theorem 1.

### 4.3 Exploiting the Donsker-Varadhan Formula

In this subsection, we instantiate the results of the previous subsections with $f(\cdot) \coloneqq \ell(\cdot, h)$, $h \in \mathcal{H}$, for some loss function $\ell \colon \mathcal{H} \times \mathcal{X} \to [0, 1]$; in this case, the results of Lemmas 3 and 5 hold for

$$Z = Z_h \coloneqq \ell(h, X), \tag{20}$$

for any hypothesis $h \in \mathcal{H}$. Next, we will need the following result which follows from the classical Donsker-Varadhan variational formula [Csiszár, 1975, Donsker and Varadhan, 1976]:

**Lemma 6.** *Let $\delta \in (0, 1)$, $\gamma > 0$ and $\rho_0$ be any fixed (prior) distribution over $\mathcal{H}$. Further, let $\{R_h : h \in \mathcal{H}\}$ be any family of random variables such that $\mathrm{E}[\exp(\gamma R_h)] \le 1$, for all $h \in \mathcal{H}$. Then, for any (posterior) distribution $\widehat{\rho}$ over $\mathcal{H}$, we have*

$$P\left[ \mathrm{E}_{h \sim \widehat{\rho}}[R_h] \le \frac{\mathrm{KL}(\widehat{\rho} \| \rho_0) + \ln \frac{1}{\delta}}{\gamma} \right] \ge 1 - \delta.$$

In addition to $Z_h$ in (20), define $Y_h \coloneqq \ell(h, X) \cdot \mathrm{E}[q_\star \mid X]$ and $Y_{h,i} \coloneqq \ell(h, X_i) \cdot \mathrm{E}[q_\star \mid X_i]$, for $i \in [n]$. Then, if we set $\gamma = \eta n$ and $R_h = \mathrm{E}_P[Y_h] - \sum_{i=1}^n Y_{h,i}/n - \eta \kappa(\eta/\alpha) \mathrm{C}_\alpha[Z_h]/\alpha$, Lemma 5 guarantees that $\mathrm{E}[\exp(\gamma R_h)] \le 1$, and so by Lemma 6 we get the following result:

**Theorem 7.** *Let $\alpha, \delta \in (0, 1)$, and $\eta \in [0, \alpha]$. Further, let $X_1, \ldots, X_n$ be i.i.d. random variables in $\mathcal{X}$. Then, for any randomized estimator $\widehat{\rho} = \widehat{\rho}(X_{1:n})$ over $\mathcal{H}$, we have, with $\widehat{Z} \coloneqq \mathrm{E}_{h \sim \widehat{\rho}}[\ell(h, X)]$,*

$$\mathrm{E}_{h \sim \widehat{\rho}}[\mathrm{C}_\alpha[\ell(h, X)]] \le \widehat{\mathrm{C}}_\alpha[\widehat{Z}](1 + \epsilon_n) + \frac{\eta \kappa(\eta/\alpha) \, \mathrm{E}_{h \sim \widehat{\rho}}[\mathrm{C}_\alpha[\ell(h, X)]]}{\alpha} + \frac{\mathrm{KL}(\widehat{\rho} \| \rho_0) + \ln \frac{1}{\delta}}{\eta n}, \tag{21}$$

*with probability at least $1 - 2\delta$ on the samples $X_1, \ldots, X_n$, where $\epsilon_n$ is defined in (11).*

If we could optimize the RHS of (21) over $\eta \in [0, \alpha]$, this would lead to our desired bound in Theorem 1 (after some rearranging). However, this is not directly possible since the optimal $\eta$ depends on the sample $X_{1:n}$ through the term $\mathrm{KL}(\widehat{\rho} \| \rho_0)$. The solution is to apply the result of Theorem 7 with a union bound, so that (21) holds for any estimator $\hat{\eta} = \hat{\eta}(X_{1:n})$ taking values in a carefully chosen grid $\mathcal{G}$; to derive our bound, we will use the grid $\mathcal{G} \coloneqq \{\alpha 2^{-1}, \ldots, \alpha 2^{-N} \mid N \coloneqq \lceil 1/2 \log_2(n/\alpha) \rceil\}$. From this point, the proof of Theorem 1 is merely a mechanical exercise of rearranging (21) and optimizing $\hat{\eta}$ over $\mathcal{G}$, and so we postpone the details to Appendix A.

## 5 New Concentration Bounds for CVAR

In this section, we show how some of the results of the previous section can be used to reduce the problem of estimating $\mathrm{C}_\alpha[Z]$ to that of estimating a standard expectation. This will then enable us to easily obtain concentration inequalities for $\widehat{\mathrm{C}}_\alpha[Z]$ even when $Z$ is unbounded. We note that previous works [Bhat and Prashanth, 2019, Kolla et al., 2019] used sophisticated techniques to deal with the unbounded case (sometimes achieving only sub-optimal rates), whereas we simply invoke existing concentration inequalities for empirical means thanks to our reduction.

The key results we will use are Lemmas 3 and 4, where we instantiate the latter with $\mathcal{X} = \mathbb{R}$ and $f \equiv \mathrm{id}$, in which case:

$$Y = Z \cdot \mathrm{E}[q_\star \mid Z], \quad \text{and} \quad q_\star \in \operatorname*{argmax}_{q \in \mathcal{Q}_\alpha} \mathrm{E}[Zq]. \tag{22}$$

Together, these two lemmas imply that, for any $\alpha, \delta \in (0, 1)$, i.i.d. random variables $Z_1, \ldots, Z_n$,

$$\mathrm{C}_\alpha[Z] - \widehat{\mathrm{C}}_\alpha[Z] - |Z_{(\lceil n\alpha \rceil)}| \epsilon_n \le \mathrm{E}[Y] - \frac{1}{n} \sum_{i=1}^n Y_i, \tag{23}$$

with probability at least $1 - \delta$, where $\epsilon_n$ is as in (11) and $Z_{(1)}, \ldots, Z_{(n)}$ are the decreasing order statistics of $Z_1, \ldots, Z_n \in \mathbb{R}$. Thus, getting a concentration inequality for $\widehat{\mathrm{C}}_\alpha[Z]$ can be reduced to getting one for the empirical mean $\sum_{i=1}^n Y_i/n$ of the i.i.d. random variables $Y_1, \ldots, Y_n$. Next, we show that whenever $Z$ is a sub-exponential [resp. sub-Gaussian] random variable, essentially so is $Y$. But first we define what this means:

**Definition 8.** *Let $\mathcal{I} \subseteq \mathbb{R}$, $b > 0$, and $Z$ be a random variable such that, for some $\sigma > 0$,*

$$\mathrm{E}[\exp\left(\eta \cdot (Z - \mathrm{E}[Z])\right)] \le \exp\left(\eta^2 \sigma^2/2\right), \quad \forall \eta \in \mathcal{I},$$

*Then, $Z$ is $(\sigma, b)$-sub-exponential [resp. $\sigma$-sub-Gaussian] if $\mathcal{I} = (-1/b, 1/b)$ [resp. $\mathcal{I} = \mathbb{R}$].*

**Lemma 9.** *Let $\sigma, b > 0$ and $\alpha \in (0, 1)$. Let $Z$ be a zero-mean real random variable and let $Y$ be as in* (22). *If $Z$ is $(\sigma, b)$-sub-exponential [resp. $\sigma$-sub-Gaussian], then*

$$\mathrm{E}[\exp(\eta Y)] \le 2\exp(\eta^2\sigma^2/(2\alpha^2)), \quad \forall \eta \in (-\alpha/b, \alpha/b) \quad [\textit{resp. } \eta \in \mathbb{R}]. \tag{24}$$

Note that in Lemma 9 we have assumed that $Z$ is a zero-mean random variable, and so we still need to do some work to derive a concentration inequality for $\widehat{\mathrm{C}}_\alpha[Z]$. In particular, we will use the fact that $\mathrm{C}_\alpha[Z - \mathrm{E}[Z]] = \mathrm{C}_\alpha[Z] - \mathrm{E}[Z]$ and $\widehat{\mathrm{C}}_\alpha[Z - \mathrm{E}[Z]] = \widehat{\mathrm{C}}_\alpha[Z] - \mathrm{E}[Z]$, which holds since $\mathrm{C}_\alpha$ and $\widehat{\mathrm{C}}_\alpha$ are coherent risk measures, and thus translation equivariant (see Definition 14). We use this in the proof of the next theorem (which is in Appendix A):

**Theorem 10.** *Let $\sigma, b > 0$, $\alpha, \delta \in (0, 1)$, and $\epsilon_n$ be as in* (11). *If $Z$ is a $\sigma$-sub-Gaussian random variable, then with $\mathrm{G}[Z] \coloneqq \mathrm{C}_\alpha[Z] - \widehat{\mathrm{C}}_\alpha[Z]$ and $t_n \coloneqq \left|Z_{(\lceil n\alpha \rceil)} - \mathrm{E}[Z]\right| \cdot \epsilon_n$, we have*

$$P\left[\mathrm{G}[Z] \ge t + t_n\right] \le \delta + 2\exp(-n\alpha^2 t^2/(2\sigma^2)), \quad \forall t \ge 0; \tag{25}$$

*otherwise, if $Z$ is $(\sigma, b)$-sub-exponential random variable, then*

$$P\left[\mathrm{G}[Z] \ge t + t_n\right] \le \delta + \begin{cases} 2\exp\left(-n\alpha^2 t^2/(2\sigma^2)\right), & \text{if } 0 \le t \le \sigma^2/(b\alpha); \\ 2\exp\left(-n\alpha t/(2b)\right), & \text{if } t > \sigma^2/(b\alpha). \end{cases}$$

We note that unlike the recent results due to Bhat and Prashanth [2019] which also deal with the unbounded case, the constants in our concentration inequalities in Theorem 10 are explicit.

When $Z$ is a $\sigma$-sub-Gaussian random variable with $\sigma > 0$, an immediate consequence of Theorem 10 is that by setting $t = \sqrt{2\sigma^2 \ln(2/\delta)/(n\alpha^2)}$ in (25), we get that, with probability at least $1 - 2\delta$,

$$\mathrm{C}_\alpha[Z] - \widehat{\mathrm{C}}_\alpha[Z] \le \frac{\sigma}{\alpha}\sqrt{\frac{2\ln\frac{1}{\delta}}{n}} + \left|Z_{(\lceil n\alpha \rceil)} - \mathrm{E}[Z]\right| \cdot \left(\sqrt{\frac{2\ln\frac{1}{\delta}}{\alpha n}} + \frac{\ln\frac{1}{\delta}}{3\alpha n}\right). \tag{26}$$

A similar inequality holds for the sub-exponential case. We note that the term $\left|Z_{(\lceil n\alpha \rceil)} - \mathrm{E}[Z]\right|$ in (26) can be further bounded from above by

$$\frac{n\alpha}{\lfloor n\alpha \rfloor}\widehat{\mathrm{C}}_\alpha[Z] - \mathrm{E}_{\widehat{P}_n}[Z] + \left|\mathrm{E}[Z] - \mathrm{E}_{\widehat{P}_n}[Z]\right|. \tag{27}$$

This follows from the triangular inequality and facts that $\widehat{\mathrm{C}}_\alpha[Z] \ge \frac{1}{n\alpha}\sum_{i=1}^{\lfloor n\alpha \rfloor} Z_{(i)} \ge \frac{\lfloor n\alpha \rfloor}{n\alpha} Z_{(\lceil n\alpha \rceil)}$ (see *e.g.* Lemma 4.1 in Brown [2007]), and $\widehat{\mathrm{C}}_\alpha[Z] \ge \mathrm{E}_{\widehat{P}_n}[Z]$ [Ahmadi-Javid, 2012]. The remaining term $\left|\mathrm{E}_P[Z] - \mathrm{E}_{\widehat{P}_n}[Z]\right|$ in (27) which depends on the unknown $P$ can be bounded from above using another concentration inequality.

Generalization bounds of the form (4) for unbounded but sub-Gaussian or sub-exponential $\ell(h, X)$, $h \in \mathcal{H}$, can be obtained using the PAC-Bayesian analysis of [McAllester, 2003, Theorem 1] and our concentration inequalities in Theorem 10. However, due to the fact that $\alpha$ is squared in the argument of the exponentials in these inequalities (which is also the case in the bounds of Bhat and Prashanth [2019], Kolla et al. [2019]) the generalization bounds obtained this way will have the $\alpha$ outside the square-root "complexity term"—unlike our bound in Theorem 1.

We conjecture that the dependence on $\alpha$ in the concentration bounds of Theorem 10 can be improved by swapping $\alpha^2$ for $\alpha$ in the argument of the exponentials; in the sub-Gaussian case, this would move $\alpha$ inside the square-root on the RHS of (26). We know that this is at least possible for bounded random variables as shown in Brown [2007], Wang and Gao [2010]. We now recover this fact by presenting a new concentration inequality for $\widehat{\mathrm{C}}_\alpha[Z]$ when $Z$ is bounded using the reduction described at the beginning of this section.

**Theorem 11.** *Let $\alpha, \delta \in (0, 1)$, and $Z_{1:n}$ be i.i.d. rvs in $[0, 1]$. Then, with probability at least $1 - 2\delta$,*

$$\mathrm{C}_\alpha[Z] - \widehat{\mathrm{C}}_\alpha[Z] \le \sqrt{\frac{12\mathrm{C}_\alpha[Z] \ln\frac{1}{\delta}}{5\alpha n}} \vee \frac{3\ln\frac{1}{\delta}}{\alpha n} + \mathrm{C}_\alpha[Z]\left(\sqrt{\frac{2\ln\frac{1}{\delta}}{\alpha n}} + \frac{\ln\frac{1}{\delta}}{3\alpha n}\right). \tag{28}$$

The proof is in Appendix A. The inequality in (28) essentially replaces the range of the random variable $Z$ typically present under the square-root in other concentration bounds [Brown, 2007, Wang and Gao, 2010] by the smaller quantity $C_\alpha[Z]$. The concentration bound (28) is not immediately useful for computational purposes since its RHS depends on $C_\alpha[Z]$. However, it is possible to rearrange this bound so that only the empirical quantity $\widehat{C}_\alpha[Z]$ appears on the RHS of (28) instead of $C_\alpha[Z]$; we provide the means to do this in Lemma 13 in the appendix.

## 6  Conclusion and Future Work

In this paper, we derived a first PAC-Bayesian bound for CVAR by reducing the task of estimating CVAR to that of merely estimating an expectation (see Section 4). This reduction then made it easy to obtain concentration inequalities for CVAR (with explicit constants) even when the random variable in question is unbounded (see Section 5).

We note that the only steps in the proof of our main bound in Theorem 1 that are specific to CVAR are Lemmas 2 and 3, and so the question is whether our overall approach can be extended to other coherent risk measures to achieve (4).

In Appendix B, we discuss how our results may be extended to a rich class of coherent risk measures known as $\varphi$-entropic risk measures. These CRMs are often used in the context of robust optimization Namkoong and Duchi [2017], and are perfect candidates to consider next in the context of this paper.

## Broader Impact

Coherent risk measures (including conditional value at risk) have been gaining significant traction in the machine learning community recently, as they allow for capturing in a much richer way the behaviour and performance of algorithms' outputs. This comes at the expense of a much harder theoretical analysis and such measures are not supported by as many guarantees than the traditional mean risk (expectation of the loss). We provide in this paper one of the few generalisation bounds for CVAR and we believe this will shed light on the advantages of using CVAR in machine learning. We intend our contributions to be of prime interest to theoreticians, but also to practitioners.

## Acknowledgments and Disclosure of Funding

This work was supported by the Australian Research Council and Data61.

## Footnotes

[1]These are precisely the properties which make coherent risk measures excellent candidates in some machine learning applications (see e.g. [Williamson and Menon, 2019] for an application to fairness)

[2]We use the convention in Brown [2007], Prashanth and Ghavamzadeh [2013], Wang and Gao [2010].

[3]The dual representation of CVAR was also leveraged in a concurrent work by Curi et al. [2020] for the purpose of efficiently optimizing CVAR objectives.

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
