[Supplementary Material]

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

# A Proofs

## A.1 Proof of Lemma 2

**Proof** Let $\varphi(\cdot) \coloneqq \iota_{[0,1/\alpha]}(\cdot)$, where for a set $\mathcal{C} \subseteq \mathbb{R}$, $\iota_{\mathcal{C}}(x) = 0$ if $x \in \mathcal{C}$; and $+\infty$ otherwise. From (12), we have that $\widetilde{C}_\alpha[Z]$ is equal to

$$\mathcal{P} \coloneqq \sup_{\boldsymbol{q}:|\mathrm{E}_{i\sim\pi}[q_i]-1|\leq\epsilon_n} \mathrm{E}_{i\sim\pi}[Z_i q_i - \varphi(q_i)], \tag{29}$$

where we recall that $\pi = (1,\ldots,1)^\top/n \in \mathbb{R}^n$. The Lagrangian dual $\mathcal{D}$ of (29) is given by

$$\mathcal{D} \coloneqq \inf_{\eta,\gamma\geq 0} \left\{ \eta - \gamma + (\eta+\gamma)\epsilon_n + \sup_{\boldsymbol{q}:0\leq q_i\leq 1/\alpha, i\in[n]} \{\mathrm{E}_{i\sim\pi}[(Z_i - \eta + \gamma)q_i - \varphi(q_i)]\} \right\},$$

$$= \inf_{\eta,\gamma\geq 0} \left\{ \eta - \gamma + (\eta+\gamma)\epsilon_n + \mathrm{E}_{i\sim\pi}\left[ \sup_{0\leq x\leq 1/\alpha} \{(Z_i - \eta + \gamma)x - \varphi(x)\} \right] \right\},$$

$$= \inf_{\eta,\gamma\geq 0} \{\eta - \gamma + (\eta+\gamma)\epsilon_n + \mathrm{E}_{i\sim\pi}[\varphi^\star(Z_i - \eta + \gamma)]\}, \tag{30}$$

$$= \inf_{\mu\in\mathbb{R}} \{\mu + |\mu|\epsilon_n + \mathrm{E}_{i\sim\pi}[\varphi^\star(Z_i - \mu)]\}, \tag{31}$$

where (30) is due to $\{x \in \mathbb{R} \mid \varphi(x) < +\infty\} = [0, 1/\alpha]$, and (31) follows by setting $\mu \coloneqq \eta - \gamma$ and noting that the inf in (30) is always attained at a point $(\eta,\gamma) \in \mathbb{R}_{\geq 0}^2$ satisfying $\eta \cdot \gamma = 0$, in which case $\eta + \gamma = |\mu|$; this is true because by the positivity of $\epsilon_n$, if $\eta, \gamma > 0$, then $(\eta+\gamma)\epsilon_n$ can always be made smaller while keeping the difference $\eta - \gamma$ fixed. Finally, since the primal problem is feasible—$\boldsymbol{q} = \pi$ is a feasible solution—there is no duality gap (see the proof of [Beck and Teboulle, 2003, Theorem 4.2]), and thus the RHS of (31) is equal to $\mathcal{P}$ in (29). The proof is concluded by noting that the Fenchel dual of $\varphi$ satisfies $\varphi^\star(z) = 0 \vee (z/\alpha)$, for all $z \in \mathbb{R}$. ∎

## A.2 Proof of Lemma 3

**Proof** Let $\widehat{\mu}$ be the $\operatorname{argmin}$ in $\mu \in \mathbb{R}$ of the RHS of (3). By Lemma 2, we have

$$\widetilde{C}_\alpha[Z] = \inf_{\mu\in\mathbb{R}} \left\{ \mu + |\mu|\epsilon_n + \frac{\mathrm{E}_{i\sim\pi}[Z_i - \mu]_+}{\alpha} \right\},$$

$$\leq \widehat{\mu} + |\widehat{\mu}|\epsilon_n + \frac{\mathrm{E}_{i\sim\pi}[Z_i - \widehat{\mu}]_+}{\alpha},$$

$$= \widehat{C}_\alpha[Z] + |\widehat{\mu}|\epsilon_n. \quad \text{(by definition of } \widehat{\mu}) \tag{32}$$

The inequality in (15) follows from (32) and the fact that $\widehat{\mu} = Z_{(\lceil n\alpha\rceil)}$ (see proof of [Brown, 2007, Proposition 4.1]).

Now we show (14) under the assumption that $Z_i \geq 0$, for all $i \in [n]$. Note that by definition $\widehat{C}_\alpha[Z] = \widehat{\mu} + \frac{1}{\alpha}\mathrm{E}_{i\sim\pi}[Z_i - \widehat{\mu}]_+$, and so $\widehat{\mu} \leq \widehat{C}_\alpha[Z]$. Furthermore, since $\alpha \in (0,1)$ and $Z_i \geq 0$, for $i \in [n]$, the RHS of (3) is a decreasing function of $\mu$ on $]-\infty, 0]$, and thus $\widehat{\mu} \geq 0$ (since $\widehat{\mu}$ is the minimizer of (3)). Combining the fact that $0 \leq \widehat{\mu} \leq \widehat{C}_\alpha[Z]$ with (32) completes the proof. ∎

## A.3 Proof of Lemma 4

**Proof** The first claim follows by the fact that $X_i, i \in [n]$, are i.i.d., and an application of the total expectation theorem. Now for the second claim, let $\Delta \coloneqq |\mathrm{E}_{\widehat{P}_n}[q_\star \mid X] - 1|$. Since $q_\star$ is a density, the total expectation theorem implies

$$\Delta = |\mathrm{E}_{\widehat{P}_n}[q_\star \mid X] - \mathrm{E}[\mathrm{E}[q_\star \mid X]]|,$$

and so by Bennett's inequality (see *e.g.* Theorem 3 in Maurer and Pontil [2009]) applied to the random variable $\mathrm{E}[q_\star \mid X]$, we get that, with probability at least $1 - \delta$,

$$
\begin{aligned}
\Delta &\le \sqrt{\frac{2\mathrm{VAR}[\mathrm{E}[q_\star \mid X]]\ln\frac{1}{\delta}}{n}} + \frac{\|\mathrm{E}[q_\star \mid X]\|_\infty \ln\frac{1}{\delta}}{3n}, \\
&\le \sqrt{\frac{2\mathrm{E}[\mathrm{E}[q_\star \mid X]^2]\ln\frac{1}{\delta}}{n}} + \frac{\|\mathrm{E}[q_\star \mid X]\|_\infty \ln\frac{1}{\delta}}{3n}, \\
&\le \sqrt{\frac{2\|\mathrm{E}[q_\star \mid X]\|_\infty \ln\frac{1}{\delta}}{n}} + \frac{\|\mathrm{E}[q_\star \mid X]\|_\infty \ln\frac{1}{\delta}}{3n},
\end{aligned}
$$

where the last inequality follows by the fact that $\mathrm{E}[\mathrm{E}[q_\star \mid X]^2] \le \mathrm{E}[\mathrm{E}[q_\star \mid X]] \cdot \|\mathrm{E}[q_\star \mid X]\|_\infty = \|\mathrm{E}[q_\star \mid X]\|_\infty$, which holds since $\mathrm{E}[q_\star \mid X] \ge 0$ and $\mathrm{E}[\mathrm{E}[q_\star \mid X]] = \mathrm{E}[q_\star] = 1$. The proof is concluded by the facts that $\|\mathrm{E}[q_\star \mid X]\|_\infty \le \|q_\star\|_\infty$ (by Jensen's inequality); $\|q\|_\infty \le 1/\alpha$, for all $q \in \mathcal{Q}_\alpha$ by definition; and $q_\star \in \mathcal{Q}_\alpha$. ∎

### A.4    Proof of Lemma 5

We need the following lemma in the proof of Lemma 5:

**Lemma 12.** *Let $S, S_1, \ldots, S_n$ be i.i.d. random variable such that $S \in [0, B]$, $B > 0$. We have,*

$$
\mathrm{E}_P\left[\exp\left(n\eta\,\mathrm{E}_P[S] - \eta\sum_{i=1}^n S_i - n\eta^2 \kappa(\eta B)\cdot \mathrm{E}_P[S^2]\right)\right] \le 1, \tag{33}
$$

*for all $\eta \in [0, 1/B]$, where $\kappa(\eta) := (e^\eta - \eta - 1)/\eta^2$.*

**Proof** The desired bound follows by the version of Bernstein's moment inequality in [Cesa-Bianchi and Lugosi, 2006, Lemma A.5] and [Mhammedi et al., 2019, Proposition 10-(b)]. ∎

**Proof of Lemma 5** By Lemma 4, the random variables $Y, Y_1, \ldots, Y_n$ are i.i.d., and so the result of Lemma 12 applies; this means that (33) holds for $(S, S_1, \ldots, S_n) = (Y, Y_1, \ldots, Y_n)$ and $B = b \ge \|Y\|_\infty$. Thus, to complete the proof it suffices to bound $\|Y\|_\infty$ and $\|Y\|_2^2 = \mathrm{E}[Y^2]$ from above. Starting with $\mathrm{E}[Y^2]$, and recalling that $Z = f(X) \in [0, 1]$ by assumption, we have:

$$
\begin{aligned}
\mathrm{E}[Y^2] &= \mathrm{E}[Z^2 \cdot \mathrm{E}[q_\star \mid X]^2], \\
&\le \mathrm{E}[Z \cdot \mathrm{E}[q_\star \mid X]] \cdot \|Z \cdot \mathrm{E}[q_\star \mid X]\|_\infty, \text{(Hölder)} \\
&\le \mathrm{C}_\alpha[Z] \cdot \|Z \cdot \mathrm{E}[q_\star \mid X]\|_\infty, \quad \text{(Lemma 4)} \\
&\le \mathrm{C}_\alpha[Z]/\alpha, \quad\quad (Z \le 1,\ q_\star \le 1/\alpha)
\end{aligned}
$$

where the fact that $q_\star \le 1/\alpha$ follows simply from $q_\star \in \mathcal{Q}_\alpha$ and the definition of $\mathcal{Q}_\alpha$. We also have

$$
\begin{aligned}
\|Y\|_\infty = \|Z \cdot \mathrm{E}[q_\star \mid X]\|_\infty &\le \|Z\|_\infty \cdot \|\mathrm{E}[q_\star \mid X]\|_\infty, \\
&\le \|q_\star\|_\infty, (Z \le 1\ \&\ \text{Jensen}) \\
&\le 1/\alpha,
\end{aligned}
$$

again the last inequality follows from $q_\star \in \mathcal{Q}_\alpha$ and the definition of $\mathcal{Q}_\alpha$. ∎

### A.5    Proof of Theorem 7

**Proof** Let $h \in \mathcal{H}$ and $\alpha, \delta \in (0, 1)$, and define

$$
R_h := \mathrm{C}_\alpha[Z_h] - \frac{1}{n}\sum_{i=1}^n Y_i - \frac{\eta\kappa(\eta/\alpha)}{\alpha}\mathrm{C}_\alpha[Z_h], \tag{34}
$$

where $Y_i := \ell(h, X_i) \cdot \mathrm{E}[q_\star \mid X_i], i \in [n]$, where $q_\star$ is as in (17) with $Z$ as in (20). By Lemma 4, $\mathrm{C}_\alpha[Z_h] = \mathrm{E}_P[Y]$, where $Y := \ell(h, X) \cdot \mathrm{E}[q_\star \mid X]$. Thus, by Lemma 5 with $Z = Z_h$, we have $\mathrm{E}_P[\exp(n\eta R_h)] \le 1$. Applying Lemma 6 with $R_h$ as in (34) and $\gamma = n\eta$, yields,

$$
\begin{aligned}
\mathrm{E}_{h\sim\widehat{\rho}}[\mathrm{C}_\alpha[\ell(h, X)]] \le\ &\frac{1}{n}\sum_{i=1}^n \ell(\widehat{\rho}, X_i) \cdot \mathrm{E}[q_\star \mid X_i] + \frac{\eta\kappa(\eta/\alpha)\,\mathrm{E}_{h\sim\widehat{\rho}}[\mathrm{C}_\alpha[\ell(h, X)]]}{\alpha} \\
&+ \frac{\mathrm{KL}(\widehat{\rho}\|\rho_0) + \ln\frac{1}{\delta}}{\eta n},
\end{aligned} \tag{35}
$$

with probability at least $1 - \delta$. Now invoking Lemmas 3 and 4 (in particular (18)), yields

$$\frac{1}{n} \sum_{i=1}^n \ell(\widehat{\rho}, X_i) \cdot \mathrm{E}[q_\star \mid X_i] \geq \widehat{\mathrm{C}}_\alpha[\widehat{Z}] \cdot (1 + \epsilon_n).$$

with probability at least $1 - \delta$, where $\widehat{Z} \coloneqq \mathrm{E}_{h \sim \widehat{\rho}}[\ell(h, X)]$. Combining this with (35) via a union bound yields the desired bound. ∎

### A.6 Proof of Theorem 1

To prove Theorem 1, we will need the following lemma:

**Lemma 13.** *Let* $R, \widehat{R}, A, B > 0$. *If* $R \leq \widehat{R} + \sqrt{RA} + B$, *then*

$$R \leq \widehat{R} + \sqrt{\widehat{R}A} + 2B + A.$$

**Proof** If $R \leq \widehat{R} + \sqrt{RA} + B$, then for all $\eta > 0$,

$$R \leq \widehat{R} + \frac{\eta}{2} R + \frac{A}{2\eta} + B,$$

which after rearranging, becomes,

$$R \leq \frac{\widehat{R}}{1 - \eta/2} + \frac{A}{2\eta \cdot (1 - \eta/2)} + \frac{B}{1 - \eta/2}, \quad \text{for } \eta \notin \{0, 2\}. \tag{36}$$

The minimizer of the RHS of (36) is given by

$$\eta = \eta_\star \coloneqq \frac{-A + \sqrt{A^2 + 4AB + 4A\widehat{R}}}{2(B + \widehat{R})}.$$

Plugging this $\eta$ into (36), yields,

$$R \leq \widehat{R} + \frac{A}{2} + B + \frac{1}{2}\sqrt{4A\widehat{R} + A^2 + 4AB},$$

$$\leq \widehat{R} + A + 2B + \sqrt{A\widehat{R}}, \tag{37}$$

where (37) follows by the facts that $A^2 + 4AB \leq (A + 2B)^2$ and $\sqrt{4\widehat{R}A + (A + 2B)^2} \leq \sqrt{4\widehat{R}A} + A + 2B$. ∎

**Proof of Theorem 1** Define the grid $\mathcal{G}$ by

$$\mathcal{G} \coloneqq \left\{ 2^{-1}\alpha, \ldots, 2^{-N}\alpha \mid N \coloneqq \lceil 1/2 \cdot \log_2 \frac{n}{\alpha} \rceil \right\},$$

and let $\hat{\eta} = \hat{\eta}(Z_{1:n}) \in \mathcal{G}$ be any estimator. Then, using the fact that $\kappa(x) \leq 3/5$, for all $x \leq 1/2$, and invoking Theorem 7 with a union bound over $\eta \in \mathcal{G}$, and $\varepsilon_n \coloneqq \sqrt{\frac{2\ln \frac{N}{\delta}}{\alpha n}} + \frac{\ln \frac{N}{\delta}}{3\alpha n}$, we get that

$$\mathrm{E}_{h \sim \widehat{\rho}}[\mathrm{C}_\alpha[\ell(h, X)]] - \widehat{\mathrm{C}}_\alpha[\widehat{Z}] \cdot (1 + \varepsilon_n) \leq \frac{\mathrm{KL}(\widehat{\rho} \| \rho_0) + \ln \frac{N}{\delta}}{\hat{\eta} n} + \frac{3\hat{\eta}}{5\alpha} \mathrm{E}_{h \sim \widehat{\rho}}[\mathrm{C}_\alpha[\ell(h, X)]], \tag{38}$$

with probability at least $1 - 2\delta$, where we recall that $\widehat{Z} = \mathrm{E}_{h \sim \widehat{\rho}}[\ell(h, X)]$. Let $\hat{\eta}$ be an estimator which satisfies

$$\hat{\eta} \in [\eta_\star \wedge (\alpha/2), \, 2\eta_\star] \cap \mathcal{G}, \quad \text{where} \quad \eta_\star \coloneqq \sqrt{\frac{5\alpha \cdot (\mathrm{KL}(\widehat{\rho} \| \rho_0) + \ln \frac{N}{\delta})}{3n \, \mathrm{E}_{h \sim \widehat{\rho}}[\mathrm{C}_\alpha[\ell(h, X)]]}} \tag{39}$$

is the unconstrained minimizer $\hat{\eta}$ of the RHS of (38). Since the loss $\ell$ has range in $[0, 1]$, $\mathrm{KL}(\widehat{\rho} \| \rho_0) \geq 0$, and $(\delta, n) \in ]0, 1/2[ \times [2, +\infty[$, we have $\eta_\star \geq \sqrt{\alpha/n} \geq \min \mathcal{G}$. This, with the fact that $\mathcal{G}$ is in the form of a geometric progression with common ratio 2 and $\max \mathcal{G} = \alpha/2$, ensures the existence (and in fact the uniqueness) of $\hat{\eta}$ satisfying (39).

**Case 1.** Suppose that $\eta_\star \le \alpha/2$. In this case, the estimator $\hat{\eta}$ in (39) satisfies $\eta_\star \le \hat{\eta} \le 2\eta_\star$. Plugging $\hat{\eta}$ into (38) yields

$$\mathrm{E}_{h\sim\widehat{\rho}}[\mathrm{C}_\alpha[\ell(h,X)]] - \widehat{\mathrm{C}}_\alpha[\widehat{Z}] \le 3\sqrt{\frac{3\,\mathrm{E}_{h\sim\widehat{\rho}}[\mathrm{C}_\alpha[\ell(h,X)]]\cdot(\mathrm{KL}(\widehat{\rho}\|\rho_0)+\ln\frac{N}{\delta})}{5\alpha n}} + \widehat{\mathrm{C}}_\alpha[\widehat{Z}]\cdot\varepsilon_n.$$

By applying Lemma 13 with $R = \mathrm{E}_{h\sim\widehat{\rho}}[\mathrm{C}_\alpha[\ell(h,X)]]$, $\widehat{R} = \widehat{\mathrm{C}}_\alpha[\widehat{Z}]$, $A = \frac{27(\mathrm{KL}(\widehat{\rho}\|\rho_0)+\ln\frac{N}{\delta})}{5\alpha n}$, and $B = \widehat{\mathrm{C}}_\alpha[\widehat{Z}]\cdot\varepsilon_n$, we get

$$\begin{aligned}
\mathrm{E}_{h\sim\widehat{\rho}}[\mathrm{C}_\alpha[\ell(h,X)]] - \widehat{\mathrm{C}}_\alpha[\widehat{Z}] \le{} & \sqrt{\frac{27\widehat{\mathrm{C}}_\alpha[\widehat{Z}]\cdot(\mathrm{KL}(\widehat{\rho}\|\rho_0)+\ln\frac{N}{\delta})}{5\alpha n}} + 2\widehat{\mathrm{C}}_\alpha[\widehat{Z}]\cdot\varepsilon_n \\
& + \frac{27(\mathrm{KL}(\widehat{\rho}\|\rho_0)+\ln\frac{N}{\delta})}{5n\alpha}.
\end{aligned} \tag{40}$$

**Case 2.** Suppose now that $\eta_\star > \alpha/2$. In this case, $\hat{\eta} = \alpha/2$. Plugging this into (38) and using the fact that $\eta_\star > \alpha/2$, yields

$$\mathrm{E}_{h\sim\widehat{\rho}}[\mathrm{C}_\alpha[\ell(h,X)]] - \widehat{\mathrm{C}}_\alpha[\widehat{Z}] \le \frac{4(\mathrm{KL}(\widehat{\rho}\|\rho_0)+\ln\frac{N}{\delta})}{\alpha n} + \widehat{\mathrm{C}}_\alpha[\widehat{Z}]\cdot\varepsilon_n. \tag{41}$$

Since $\widehat{\mathrm{C}}_\alpha[\widehat{Z}] \ge 0$ and $4 \le 27/5$, the RHS of (41) is less than the RHS of (40), which completes the proof. ∎

## A.7 Proof of Lemma 9

**Proof** Suppose that $Z$ is $(\sigma, b)$-sub-exponential. Then,

$$\mathrm{E}[e^{\eta Z}] \le e^{\frac{\eta^2\sigma^2}{2}}, \quad \forall |\eta| \le 1/b. \tag{42}$$

Using that $\mathrm{E}[q_\star \mid Z] \le 1/\alpha$, we get

$$|\eta Y| \le |\eta Z|/\alpha, \quad \forall \eta \in \mathbb{R}, \tag{43}$$

and so, for all $|\eta| \le \alpha/b$, we have

$$\mathrm{E}[e^{\eta Y}] \le \mathrm{E}[e^{|\eta Y|}] \overset{(43)}{\le} \mathrm{E}[e^{\frac{\eta Z}{\alpha}}] + \mathrm{E}[e^{-\frac{\eta Z}{\alpha}}] \overset{(42)}{\le} 2e^{\frac{\eta^2\sigma^2}{2\alpha^2}}.$$

When $Z$ is $\sigma$-sub-Gaussian case, the proof is the same, except that we replace $b$ by $0$. ∎

## A.8 Proof of Theorem 11

**Proof** Let $\mathcal{X} = [0,1]$ and $f \equiv \mathrm{id}$ be the identity map. By invoking Lemmas 3 and 5 with $Z = f(X) = X$; and using (19) (which is a consequence of Lemma 4), we get, for all $\eta \in [0,\alpha]$,

$$\mathrm{E}_P\left[\exp\left(n\eta\cdot\left(\mathrm{C}_\alpha[Z] - \widehat{\mathrm{C}}_\alpha[Z](1+\epsilon_n) - \frac{\eta\kappa(\eta/\alpha)\mathrm{C}_\alpha[Z]}{\alpha}\right)\right)\right] \le 1, \tag{44}$$

with probability at least $1-\delta$, where $\epsilon_n$ is as in (11). By adding $\mathrm{C}_\alpha[Z]\cdot\epsilon_n$ to both sides of (44) and using the fact that $\kappa(x) \le 3/5$, for all $x \le 1/2$, we get, for all $\eta \in [0,\alpha/2]$,

$$\mathrm{E}_P\left[\exp\left(n\eta\cdot\left(\mathrm{C}_\alpha[Z] - \widehat{\mathrm{C}}_\alpha[Z] - \left(\frac{3\eta}{5\alpha}+\epsilon_n\right)\mathrm{C}_\alpha[Z]\right)\right)\right] \le 1, \tag{45}$$

with probability at least $1-\delta$. Let $W \coloneqq \mathrm{C}_\alpha[Z] - \widehat{\mathrm{C}}_\alpha[Z] - \left(\frac{3\eta}{5\alpha}+\epsilon_n\right)\mathrm{C}_\alpha[Z]$, and note that by (45), we have

$$P[\mathrm{E}_P[\exp(n\eta W)] \le 1] \ge 1-\delta.$$

Let $\mathcal{E}$ be the event that $\mathrm{E}_P[\exp(n\eta W)] \le 1$. With this, we have, for any $\delta \in (0,1)$ and all $\eta \in [0, \alpha/2]$,

$$P\left[\mathrm{C}_\alpha[Z] - \widehat{\mathrm{C}}_\alpha[Z] \ge \left(\frac{3\eta}{5\alpha} + \epsilon_n\right)\mathrm{C}_\alpha[Z] + \frac{\ln\frac{1}{\delta}}{\eta n}\right] = P\left[e^{n\eta W} \ge \frac{1}{\delta}\right]$$

$$= P\left[e^{n\eta W} \ge \frac{1}{\delta}\Big|\mathcal{E}\right] \cdot P[\mathcal{E}]$$

$$+ P\left[e^{n\eta W} \ge \frac{1}{\delta}\Big|\mathcal{E}^{\mathrm{c}}\right] \cdot (1 - P[\mathcal{E}]),$$

$$\le \delta\,\mathrm{E}[e^{n\eta W}\mid\mathcal{E}] + \delta, \tag{46}$$

$$\le 2\delta, \qquad \text{(by definition of } \mathcal{E}) \tag{47}$$

where (46) follows by Markov's inequality and (45). Now, we can re-express (47) as

$$\mathrm{C}_\alpha[Z] - \widehat{\mathrm{C}}_\alpha[Z] \le \left(\frac{3\eta}{5\alpha} + \epsilon_n\right)\mathrm{C}_\alpha[Z] + \frac{\ln\frac{1}{\delta}}{\eta n},$$

with probability at least $1 - 2\delta$. By setting $\eta = \sqrt{\frac{5\alpha\ln\frac{1}{\delta}}{3n\mathrm{C}_\alpha[Z]}} \wedge \alpha/2$ (which does not depend on the samples), we get

$$\mathrm{C}_\alpha[Z] - \widehat{\mathrm{C}}_\alpha[Z] \le \epsilon_n\mathrm{C}_\alpha[Z] + \sqrt{\frac{12\mathrm{C}_\alpha[Z]\ln\frac{1}{\delta}}{5\alpha n}} \vee \frac{3\ln\frac{1}{\delta}}{\alpha n},$$

with probability at least $1 - 2\delta$. ∎

### A.9 Proof of Theorem 10

**Proof** Let $\bar{Z} = Z - \mathrm{E}[Z]$. Suppose that $Z$ is $(\sigma, b)$-sub-exponential. In this case, by Lemma 9 the random variable $Y := \bar{Z} \cdot \mathrm{E}[q_\star \mid \bar{Z}]$ satisfies (24), and so by [Wainwright, 2019, Theorem 2.19], we have

$$P\left[\mathrm{E}[Y] - \frac{1}{n}\sum_{i=1}^n Y_i \ge t\right] \le \begin{cases} 2e^{-\frac{n\alpha^2 t^2}{2\sigma^2}}, & \text{if } 0 \le t \le \frac{\sigma^2}{b\alpha}; \\ 2e^{-\frac{n\alpha t}{2b}}, & \text{if } t > \frac{\sigma^2}{b\alpha}. \end{cases} \tag{48}$$

For any real random variables $A, B$, and $C$, we have $[A \ge C] \implies [A \ge B \text{ or } B \ge C]$, and so $P[A \ge C] \le P[A \ge B] + P[B \ge C]$. Applying this with $A = \mathrm{C}_\alpha[\bar{Z}] - \widehat{\mathrm{C}}_\alpha[\bar{Z}] - |\bar{Z}_{(\lceil n\alpha\rceil)}|\epsilon_n$, $B = \mathrm{E}[Y] - \sum_{i=1}^n Y_i/n$, and $C = t \in \mathbb{R}$. we get:

$$P\left[\mathrm{C}_\alpha[\bar{Z}] - \widehat{\mathrm{C}}_\alpha[\bar{Z}] - |\bar{Z}_{(\lceil n\alpha\rceil)}|\cdot\epsilon_n \ge t\right] \le P\left[\mathrm{C}_\alpha[\bar{Z}] - \widehat{\mathrm{C}}_\alpha[\bar{Z}] - |\bar{Z}_{(\lceil n\alpha\rceil)}|\cdot\epsilon_n \ge \mathrm{E}[Y] - \frac{1}{n}\sum_{i=1}^n Y_i\right]$$

$$+ P\left[\mathrm{E}[Y] - \frac{1}{n}\sum_{i=1}^n Y_i \ge t\right],$$

$$\le \delta + \begin{cases} 2e^{-\frac{n\alpha^2 t^2}{2\sigma^2}}, & \text{if } 0 \le t \le \frac{\sigma^2}{b\alpha}; \\ 2e^{-\frac{n\alpha t}{2b}}, & \text{if } t > \frac{\sigma^2}{b\alpha}, \end{cases} \tag{49}$$

where the last inequality follows by (48) and the fact that (23) (with $Z$ replaced by $\bar{Z}$) holds with probability at least $1 - \delta$. Since $\mathrm{C}_\alpha[Z]$ [resp. $\widehat{\mathrm{C}}_\alpha[Z]$] is a coherent risk measure, we have $\mathrm{C}_\alpha[\bar{Z}] = \mathrm{C}_\alpha[Z] - \mathrm{E}[Z]$ [resp. $\widehat{\mathrm{C}}_\alpha[\bar{Z}] = \widehat{\mathrm{C}}_\alpha[Z] - \mathrm{E}[Z]$], and so the LHS of (49) is equal to

$$P\left[\mathrm{C}_\alpha[Z] - \widehat{\mathrm{C}}_\alpha[Z] \ge t + |\bar{Z}_{(\lceil n\alpha\rceil)}|\cdot\epsilon_n\right].$$

This with the fact that $\bar{Z}_{(\lceil n\alpha\rceil)} = Z_{(\lceil n\alpha\rceil)} - \mathrm{E}[Z]$ completes the proof for the sub-exponential case.

When $Z$ is $\sigma$-sub-Gaussian case, the proof is the same, except that we replace $b$ by $0$ and use the convention that $0/0 = +\infty$. ∎

# B  Beyond CVaR

First, we give a formal definition of a coherent risk measure (CRM):

**Definition 14.** *We say that* $R \colon \mathcal{L}^1(\Omega) \to \mathbb{R} \cup \{+\infty\}$ *is a coherent risk measure if, for any* $Z, Z' \in \mathcal{L}^1(\Omega)$ *and* $c \in \mathbb{R}$*, it satisfies the following axioms: (Positive Homogeneity)* $R[\lambda Z] = \lambda R[Z]$*, for all* $\lambda \in (0,1)$*; (Monotonicity)* $R[Z] \le R[Z']$ *if* $Z \le Z'$ *a.s.; (Translation Equivariance)* $R[Z + c] = R[Z] + c$*; (Sub-additivity)* $R[Z + Z'] \le R[Z] + R[Z']$*.*

It is known that the conditional value at risk is a member of a class of CRMs called $\varphi$-entropic risk measures Ahmadi-Javid [2012]. These CRMs are often used in the context of robust optimization Namkoong and Duchi [2017], and are perfect candidates to consider next in the context of this paper:

**Definition 15.** *Let* $\varphi \colon [0, +\infty[ \to \mathbb{R} \cup \{+\infty\}$ *be a closed convex function such that* $\varphi(1) = 0$*. The* $\varphi$*-entropic risk measure with divergence level* $c$ *is defined as*

$$\mathrm{ER}_{\varphi}^c[Z] \coloneqq \sup_{q \in \mathcal{Q}_{\varphi}^c} \mathrm{E}_P[Zq], \ \ where$$

$$\mathcal{Q}_{\varphi}^c \coloneqq \left\{ q \in \mathcal{L}^1(\Omega) \,\middle|\, \begin{matrix} \exists Q \in \mathcal{M}_P(\Omega), q = \mathrm{d}Q/\mathrm{d}P, \\ \mathrm{D}_{\varphi}(Q\|P) \le c \end{matrix} \right\},$$

*and* $\mathrm{D}_{\varphi}(Q\|P) \coloneqq \mathrm{E}_P[\varphi(q)]$ *is the* $\varphi$*-divergence between two distributions* $Q$ *and* $P$*, where* $Q \ll P$ *and* $q = \frac{\mathrm{d}Q}{\mathrm{d}P}$*.*

As mentioned above, $\mathrm{CVaR}_{\alpha}[Z]$ is a $\varphi$-entropic risk measure; in fact, it is the $\varphi$-entropic risk measure at level $c = 0$ with $\varphi(\cdot) \coloneqq \iota_{[0,1/\alpha]}(\cdot)$, where for a set $\mathcal{C} \subseteq \mathbb{R}$, $\iota_{\mathcal{C}}(x) = 0$ if $x \in \mathcal{C}$; and $+\infty$ otherwise Ahmadi-Javid [2012].

The natural estimator $\widehat{\mathrm{ER}}_{\varphi}^c[Z]$ of $\mathrm{ER}_{\varphi}^c[Z]$ is defined by Ahmadi-Javid [2012]

$$\widehat{\mathrm{ER}}_{\varphi}^c[Z] = \inf_{\nu > 0, \mu \in \mathbb{R}} \left\{ \mu + \nu \, \mathrm{E}_{\widehat{P}_n} \left[ \varphi^{\star} \left( \frac{Z - \mu}{\nu} - c \right) \right] \right\}.$$

Extending the results of Lemmas 2 and 3 comes down to finding an auxiliary estimator $\widetilde{\mathrm{ER}}_{\varphi}^c[Z]$ of $\mathrm{ER}_{\varphi}^c[Z]$ which satisfies (as in Lemma 3) $\widetilde{\mathrm{ER}}_{\varphi}^c[Z] \le \widehat{\mathrm{ER}}_{\varphi}^c[Z] \cdot (1 + \epsilon_n)$, for some "small" $\epsilon_n$, and

$$\frac{1}{n} \sum_{i=1}^n Z_i \cdot \mathrm{E}[q_{\star} \mid Z_i] \le \widetilde{\mathrm{ER}}_{\varphi}^c[Z],$$

with high probability, where $q_{\star} \in \mathrm{argmin}_{q \in \mathcal{Q}_{\varphi}^c} \mathrm{E}[Zq]$. The similarities between the expressions of $\widehat{\mathrm{ER}}_{\varphi}^c[Z]$ and $\widehat{\mathrm{C}}_{\alpha}[Z]$ hint that it might be possible to find such an estimator by carefully constructing a set $\widetilde{\mathcal{Q}}_{\varphi}^c$ to play the role of the $\widetilde{\mathcal{Q}}_{\alpha}$ in Section 4. We leave such investigations for future work.