[Reviews · NeurIPS 2020]

Review 1

Summary and Contributions: CVaR (Conditional Value at Risk or Expected Shortfall) of a random variable is the expected value of this random variable conditionally on the fact that this random variable exceeds a given quantile (the quantile is the Value at Risk). CVaR has gained popularity in quantitative risk management as CVaR is a coherent risk measure (CRM) while VaR is not. In recent years, Machine Learners have explored using CVaR as an alternative to average loss. This motivates minimizing an Empirical CVaR as a proxy for minimizing CVaR. Justifying this version of Empirical Risk Minimization requires controlling the fluctuations of the Empirical CVaR around CVaR. The paper develops PAC-Bayesian bounds for CVaR (Theorem 1): it provides a high probability upper bound on the CVaR of hypotheses picked according to a posterior distribution in terms of empirical CVaR of this posterior and of the relative entropy between the posterior and a fixed prior (this has the form of other PAC-Bayesian bounds). The proof (which is non-trivial in its current form) relies on several twists. The first one is a variational formulation of Coherent Risk Measures. Then the authors introduce a sequence of intermediate random variables that can be compared with empirical CVaR and that are amenable to stochastic analysis. It is not clear to me whether this approach can be simplified and made more transparent, but this manuscript represents a substantial body of work.

Strengths: - Explore an interesting risk criterion - Non trivial concentration for empirical risk - Nice flops between variational characterizations

Weaknesses: - Generalization bounds are not as interesting as excess risk bounds - Few comments on computational hardness of the CVaR minimization problem (do we need to consider surrogate risks) - More perspective on the proof techniques could help

Correctness: So far so good.

Clarity: Writing could be improved in two respects: build stronger motivation for investigating CVaR; make the derivation of the PAC-Bayesian bound more transparent. Bayesian (even PAC-Bayesian) are not always edible. Make readers life easier.

Relation to Prior Work: Yes

Reproducibility: Yes

Additional Feedback:


Review 2

Summary and Contributions: This paper introduces a PAC-Bayesian generalization bound for the Conditional Value at Risk, CVaR, performance measure. On the way, they establish a new concentration inequality for the CVaR.

Strengths: Strengths: - Solid technical paper focusing on a risk measure that is relevant for real-life scenario, in the PAC-Bayesian framework. - A smart solution to get the bound, using an auxiliary estimator of the CVaR, properties of Coherent Risk Measures, and the folk Donsker-Varadhan Formula. - A concentration bound for the CVaR, which has some value on its own.

Weaknesses: Weaknesses: generally speaking, a broader perspective is lacking - Starting with the concentration bound of section 5, isn't it possible to get a wider range of generalization bounds, i.e bounds from other frameworks other than the PAC-Bayes framework such as, eg, Rademacher bounds? This part is not discussed. - Also, in the same vein: concentration inequalities are essential to derive generalization bounds and it's even "straightforward" to get PAC Bayes bounds from concentration inequalities. It is not discussed how this route could have been taken for the paper: 1) introduce the concentration result and 2) give the PAC Bayes bound. - Coherent risk measures are introduced and a bit discussed, but it is not clear how the results provided here cannot carry over to those CRMs in general, the result for CVaR being a specific case.

Correctness: From a mathematical point of view: everything is correct. No empirical results provided: the authors could have given a hint on how practical the provided results could become.

Clarity: The paper is clearly written. Again, my main feedback is about the perspective: a broader perspective could have implied a totally different write-up with yet more clarity -- and/but that would be a significantly different paper.

Relation to Prior Work: A clear statement is made as how the present work differs from previous work: proof techniques, tightness of the result, possible unboundedness of the risk considered.

Reproducibility: Yes

Additional Feedback: - a(n) hypothesis - Line 294: should be Lemma 13 instead of Lemma 12.


Review 3

Summary and Contributions: This paper proposes a general framework for machine learning applications based on the minimization of an empirical estimate of the conditional value at risk (CVaR). The main goal of this article is to provide theoretical guarantees for the excess of risk. Although this topic has been extensively studied in the literature for expected loss, the case of the CVaR is not as well understood. The authors start by recalling existing guarantees for the CVaR. Then they present their PAC-Bayesian generalization bound for the CVaR, which is tight in the parameter alpha, contrary to previous analysis. Finally, they describe the proof, which relies on a reduction to the expected loss for an auxiliary random variable Y.

Strengths: Strong theoretical contribution. May be used for risk-aware methods based on the conditional value at risk.

Weaknesses: Illustrations, both experimental and theoretical, are missing. For instance, what is the optimal classifier in binary classification with 0-1 loss in the CVaR case?

Correctness: Yes. The ERM with CVaR framework is correctly defined, and then analysed.

Clarity: The paper is very well written.

Relation to Prior Work: Yes, related works are discussed and existing results are recalled.

Reproducibility: Yes

Additional Feedback: Question 1: What is the optimal classifier in binary classification with the CVaR of the 0-1 loss? Is it a variant of the Bayes classifier with the auxiliary random variable Y (from the reduction to expectation)? --> answered in the rebuttal: it is the “new” SVM Question 2: More practically, could this reduction to expectation trick be used for optimization purpose, by e.g. using the Stochastic Gradient Descent algorithm? --> this reduction is for now a theoretical trick: not straightforward to leverage it for optimization purpose


Review 4

Summary and Contributions: This paper provides statistical insights in learning with Conditional Value at Risk (CVaR). It provides concentration inequalities for CVaR when the loss function is bounded or unbounded (resulting in a sub-exponential random variable). In the bounded case, the authors derive the main result of the paper, which is a sharp PAC-Bayesion generalization bound for learning with CVaR.

Strengths: The contribution of this manuscript is theoretical and definitely relevant for the NeurIPS community. As far as I know, the results are novel (sharper than previous ones) and sounds. The paper is very well written and presents pedigagically a technical work (I appreciate this initiative greatly and I thank the authors for it). This is a valuable work.

Weaknesses: As far as I am concerned, the only flaws of the paper are three typos: - Line 23: "uniform converges arguments" may mean "uniforme convergence arguments". - Line 254: exp is missing in the lhs. - Line 303: "Appendix E" refers actually to Appendix B (LaTeX compilation bug).

Correctness: Everything seems correct.

Clarity: The paper is very clear.

Relation to Prior Work: Relation to prior work is adequately addressed.

Reproducibility: Yes

Additional Feedback: I thank the authors for addressing my comments in their rebuttal.

[Author Response · NeurIPS 2020]

We warmly thank all four reviewers for their careful reading and evaluation of our work, and for their input which helped improve the manuscript. We very much appreciate the overall positive assessment from all reviewers. We provide now individual responses to some of the questions or comments in the reviews.

**Reviewer #1.** *"Few comments on computational hardness of the CVaR minimization problem (do we need to consider surrogate risks"* Minimizing the empirical CVaR is not much harder than minimizing the standard empirical expectation. In fact, the expression of the empirical CVaR in display (3) reveals that one can reduce the problem of minimizing CVaR to that of minimizing an empirical average with an extra real variable—this is the $\mu$ variable in display (3).

*"build stronger motivation for investigating CVaR; make the derivation of the PAC-Bayesian bound more transparent. Bayesian (even PAC-Bayesian) are not always edible. Make readers life easier."* With the extra space provided by the ninth page, we will add more on the motivation behind CVaR and some additional (proof) details in Subsection 4.3 (i.e. the last step in deriving the PAC-Bayesian bound).

**Reviewer #2.** *" Starting with the concentration bound of section 5, isn't it possible to get a wider range of generalization bounds, i.e bounds from other frameworks other than the PAC-Bayes framework such as, eg, Rademacher bounds? This part is not discussed."*

*"Also, in the same vein... 1) introduce the concentration result and 2) give the PAC Bayes bound."*

Starting from concentration inequalities (such as the one in Theorem 11), it is certainly possible to recover generalization bounds either through Rademacher analysis or via the PAC-Bayesian analysis due to McAllester (we discuss the latter possibility in the two paragraphs between the lines 145 and 161). However, starting from Theorem 11, for example, and directly applying such techniques, will yield looser bounds than the one we present in our main Theorem 1 (in the best case, some terms will be off by a "Jensen gap"; an instance of this is described in lines 152 and 153). We avoid this gap in Theorem 1 because we use a bound on the moment generating function of the auxiliary random variable $Y$—this is Lemma 5. This lemma is stronger than Theorem 11 (in fact, Lemma 5 implies Theorem 11), and so the former leads to a tighter generalization bound. However, we did not want Lemma 5 to take center stage in the story since it is less interpretable; it involves the *implicit* auxiliary random variable $Y$. We will add a note on this matter in the final version.

*"Coherent risk measures are introduced and a bit discussed, but it is not clear how the results provided here cannot carry over to those CRMs in general, the result for CVaR being a specific case."*

Our technique relies on a dual property of CVaR, which is not necessarily shared by all CRMs. In particular, there exists a convex function $\varphi$ such that $\mathrm{CVaR}[X] = \sup_{Q \in \mathcal{B}_\varphi} \mathbb{E}_Q[X]$, where $\mathcal{B}_\varphi$ is a $\varphi$-divergence ball. Varying the choice of the convex function $\varphi$ leads to a rich class of CRMs called *entropic risk measures*. In Appendix B, we explain how our techniques may be transferred to this case, given the structural similarity. However, it is not clear to us how to obtain generalization bounds for all CRMs beyond entropic risk measures. We consider this an exciting direction for future research.

**Reviewer #3.** *"Question 1: What is the optimal classifier in binary classification with the CVaR of the 0-1 loss? Is it a variant of the Bayes classifier with the auxiliary random variable Y (from the reduction to expectation)?"*

The optimal classifier in binary classification with CVaR of the 0-1 loss is the "new" SVM—see e.g. the paper "$\nu$-support vector machine as conditional value-at-risk minimization" by Takeda and Masashi 2008.

*"Question 2: More practically, could this reduction to expectation trick be used for optimization purpose, by e.g. using the Stochastic Gradient Descent algorithm?"* The reduction to the expected risk is only useful in the analysis; note that the random variable $Y$ (as in display (17)) that we introduce in the reduction depends on the "support point" $q_\star$ in the dual formulation of CVaR. This support point is implicit (it depends on the unknown data-generating distribution), and so it is not clear how it can be used for practical optimization purposes.

**Reviewer #4.** We thank the reviewer for their feedback. We will correct the typos which were identified.

[Meta-Review · NeurIPS 2020]

Conditional Value at Risk or Expected Shortfall (CVaR) of a random variable is the expected value of this random variable conditionally on the fact that this random variable exceeds a given value. As example, it quantifies the amount of tail risk an investment portfolio has. This kind of value is of importance in many situations and is getting more attention in the ML community. Indeed, a learned predictor that has a bad accuracy might nevertheless be of high utility if it get some a high CVaR provided there is a particular interest for examples that are in the best quantile (e.g., best drivers for a car insurance compagnies ... the only ones that should qualify for a reduction of their insurance quotes. There is still a lot to understand on CVaR from the learning theory point of view, this paper proposes the first known PAC-Bayesian bound for CVaR. This results has to be shared with the NeurIPS community.